# Chronic Effects of Different Types of Neuromuscular Training on Hemodynamic Responses Estimated VO_2max_, and Walking Performance in Older People

**DOI:** 10.3390/ijerph20010640

**Published:** 2022-12-30

**Authors:** Estélio Henrique Martin Dantas, Leandro de Oliveira Sant’Ana, Jeferson Macedo Vianna, Sergio Machado, Jani Cleria Pereira Bezerra, Matthew T. Corey, Fabiana Rodrigues Scartoni

**Affiliations:** 1Stricto Sensu Graduate Program in Nursing and Biosciences—PPgEnfBio, Federal University of Rio de Janeiro—UNIRIO, Rio de Janeiro 21941-901, Brazil; 2Master and Doctor’s Degree Program in Health and Environment—PSA, Tiradentes University—UNIT, Aracaju 49032-390, Brazil; 3Postgraduate Program in Physical Education, Federal University of Juiz de Fora, Juiz de Fora 36036-900, Brazil; 4Strength Training Studies and Research Laboratory, Federal University of Juiz de Fora, Juiz de Fora 36036-900, Brazil; 5Laboratory of Physical Activity Neuroscience, Neurodiversity Institute, Queimados 26325-020, Brazil; 6Department of Sports Methods and Techniques, Federal University of Santa Maria, Santa Maria 97105-900, Brazil; 7Campo Grande, University Center of Rio de Janeiro, Rio de Janeiro 21941-901, Brazil; 8Harvard Medical School, Boston, MA 02115, USA; 9Cambridge Health Alliance, Cambridge, MA 02143, USA; 10Sport and Exercise Sciences Laboratory, Catholic University of Petrópolis, Petrópolis 25685-100, Brazil

**Keywords:** resistance training, hemodynamics, VO_2max_, health, elderly

## Abstract

This paper investigated the effects over time of different forms of neuromuscular training on hemodynamic responses, the estimated VO_2max_, and walking performance. 105 older adults were randomly organized into three groups: RG_A_, RG_B_, and the Control Group (CG). RG_A_ and RG_B_ did 4 weeks of adaptation phase training and 12 weeks of intervention with different loads: moderate loads for RG_B_. and higher loads for RG_A_. A pre- and post-evaluation of the resting heart rate (HR), systolic blood pressure (SBP), diastolic blood pressure (DBP), double product (DP), estimated VO_2max_, and walking performance were assessed. Significant differences were observed for SBP, DBP, HR, and DP. For SBP, a post-evaluation reduction was observed only in RG_A_ (*p* = 0.007) and when comparing RG_A_ with the Control Group (*p* < 0.000). For the absolute VO_2max,_ a significant improvement was seen in RG_B_ compared to RG_A_ (*p* = 0.037) and CG (*p* < 0.000). For the relative VO_2max,_ RG_B_ scored significantly higher than RG_A_ (*p* < 0.000) and CG (*p* < 0.000), post-intervention. For the walk test, a significant reduction in completion times was observed for RG_A_ (*p* = 0.027) and RG_B_ (*p* < 0.000), and for RG_B_ compared to RG_A_ (*p* = 0.000) and CG (*p* < 0.000). Resistance training can be an excellent strategy for hemodynamic and cardiorespiratory improvement in the elderly.

## 1. Introduction

Aging brings on several organic changes in the physical and functional capacities of the elderly. Whether cardiovascular [1], cardiorespiratory [2], neuromuscular [3,4], or neurological [5], these changes can make physical and functional capacities less efficient in older adults [6]. There is evidence that physical and functional declines in the elderly are more closely related to low physical conditioning than age itself [7]. Fortunately, therefore, some of these changes can be reversed. Improvements in the physical and functional capacity of the elderly, through focused physical training, have been shown to benefit quality of life [8,9].

The domain most affected by aging is neuromuscular capacity, which can eventually lead to sarcopenia [9]. Physical exercise is already well established in the literature as a primary intervention for the elderly [10] to combat neuromuscular decline [11]. Specific protocols for neuromuscular training in the elderly have been published, highlighting several essential aspects for this class of activity [12]. Alongside the neuromuscular, other physiological systems also require thorough attention in the elderly, such as the cardiovascular and cardiorespiratory systems, and physical training is key to promoting better conditioning across these domains [13,14], especially in the older adult population [10,11].

There is, however, little research on the effect of neuromuscular training, commonly known as resistance training, on cardiovascular and cardiorespiratory responses in the elderly [12,15]. We asked how neuromuscular training can influence cardiorespiratory fitness since this is an essential aspect of health and will often decline as part of the aging process [16,17]. Our study highlights the chronic effects of different neuromuscular training protocols on hemodynamic responses, the estimated VO_2 max_, and walking performance in older adult subjects.

## 2. Materials and Methods

The study included 105 older adults (64 women and 41 men) aged 60 or over (Table 1). As inclusion criteria, each individual was required to (1) be fit to participate in the experimental intervention with no osteo-articular restriction, (2) be independent, and (3) no do regular physical exercise for at least three months. As exclusion criteria, the multidisciplinary team excluded subjects with any intervention restrictions (such as musculoskeletal injuries) and subjects using drugs to control and/or balance the cardiovascular system for psychological or neurological indications. Finally, other factors that could negatively affect the intervention results, such as morbid obesity and chronic kidney disease, were considered exclusion criteria.

The present study met the standards for research on human beings, following resolution 466/12, the National Health Council of 12 December 2012 (Brazilian Ministry of Health), and the Helsinki Resolution [18,19]. The study submitted its research project to the Research Ethics Committee Involving Human Beings at Castelo Branco University. All participants agreed to sign the informed consent form affirming the study objective, assessment procedures, and the voluntary nature of the subject’s participation. A consent form was also prepared for the institution where the research was carried out, with the same items as the informed consent form and the questionnaire for physical activity readiness (PAR-Q).

### 2.1. Experimental Design

Subjects were randomly organized into three groups, namely: Resistance Group A (RG_A_, n = 35), Resistance Group B (RG_B_, n = 35), and Control Group (CG, n = 35). Before starting training, all individuals made three visits to the laboratory. The first was to become familiar with all procedures, to sign and fill out documents (informed consent form and PAR-Q), and to submit to an anthropometric, hemodynamic, and cardiopulmonary evaluation. The second and third visits (48-h intervals) were for the test and retest of the weight test of one repetition maximum (1RM) to outline the loads applied in the training program exercises. To measure muscle strength, 1RM was measured for the following exercises: bench press, simultaneous squat, one-arm dumbbell row, seated knee extension, simultaneous standing biceps, standing knee flexion, and triceps on the bench.

Training continued for 16 weeks (Figure 1), with two sessions per week, with breaks between 48 and 72 h between sessions. The alternating method by segment was used for training sessions, and the following exercises were selected: bench press, simultaneous squat, one-arm dumbbell row, seated knee extension, simultaneous standing biceps, standing knee flexion, and triceps on the bench. All exercises were performed freely (without aid from an apparatus). The Control Group (CG) maintained routine daily tasks throughout the study. In addition, this group performed no systematic physical activity during the 16 weeks. Once a week, the Control Group met with the evaluators to report their daily activity routine for the week. The experimental groups performed the intervention until the post-test and performed the same evaluations as the experimental groups in the same period, with measurements collected pre- and post-intervention. For the experimental groups, the training sessions followed periodic (4 weeks) and specific (12 weeks) adaptation phases [20]:

### 2.2. Training Protocols

Both experimental groups (RG_A_ and RG_B_) performed 4 weeks of adaptation and 12 weeks of specific training load with 70–85% and 50–70% of 1RM, respectively. RG_A_ and RG_B_ conducted two weekly sessions with a resting interval of between 48 and 72 h. The same exercise program was used in the experimental groups, using the alternating method by segment with the bench press, squat, one-sided curved stroke, leg extension, biceps curl, knee flexion, and triceps forehead.

Both phases had 10 minutes of warm-up, with exercises geared toward major joint mobility. At the end of each session, stretching exercises lasting 5 minutes were performed for muscle relaxation [21]. Participants were instructed to breathe with each repetition, exhaling in the concentric phase of the movement and inhaling in the eccentric phase, as is the most suitable for this age group. Each group performed a series of repetitions with varying interval time, execution speed, and workload (Figure 2). The Control Group performed only routine daily activities (without excessive effort) and evaluations in the same experimental minigroups.

### 2.3. Anthropometric Analyses

Body composition was measured using two parameters: body mass index and left calf circumference [20]. Body mass and height were measured to verify the Body Mass Index, which is essential for assessing the older adult’s nutritional and anthropometric profile [22].

In the protocol for measuring body mass and height, the participant is weighed barefoot and in exercise clothes (light clothes, shorts, and shirt), in a standing position, in the central part of the platform of the Filizola^®^ brand mechanical scale (Brazil) with INMETRO seal, accurate to 100 g, with weight measured in kilograms.

To measure height, the same standard mentioned above was used, with an aluminum stadiometer, with 1 mm precision, standing upright, arms extended along the body, feet together, in inspiratory apnea, with the head oriented in the Frankfurt plane. Height was measured in centimeters [23].

Then, measurements of body mass and height were used to calculate the body mass index (BMI), obtained by the ratio between the measurements expressed in the formula below, with body mass being measured in kilograms (kg) and height measured in meters (m): BMI = Body mass/height in m^2^.

### 2.4. Hemodynamic Analyses

The variables of resting heart rate (HR) and blood pressure, systolic and diastolic, were used for hemodynamic analysis. Also, the double product was computed using the equation: [HR (bpm) × SBP (mm/Hg)] [24]. A POLAR RS800CX^®^ watch (Multisport ™ model), Kempele, Finland^®^, [24] was used for HR collection. For the assessment of systolic and diastolic blood pressure (SBP and DBP, respectively), an OMRON M6 (HEM-7001- E)^®^ digital oscillometer was used [25]. After 5 min of rest, the individuals were seated with their arms extended and placed at the heart level [26].

### 2.5. Maximum Oxygen Uptake Analyses (VO_2max_) and Walking Performance

The cardiorespiratory analysis was performed using the 1600-m test [27], one of the most recommended assessments for the elderly population [28]. The test was conducted on a soccer field with official measures, and the objective was for the participants to complete the course in the shortest possible time. HR was collected immediately after the test. Time was measured using an OREGON^®^ stopwatch (model C510-B™). Heart rate was collected using a POLAR RS800CX^®^ watch (Multisport model ™), from Kempele, Finland [29]. After collecting the variables (time and HR), the equation below was used to calculate the estimated value of VO_2max_. The duration of the activity was performed to assess the performance of the walk.

Equation:VO_2max_ = 132.853 − (0.0769 × Bodyweight) − (0.3877 × age) + (6.315 × gender) − (3.2649 × time) − (0.01565 × HR)
where lb (1 lb = 0.454); gender: female = 0/male = 1; time: minutes; HR: collected after test.

### 2.6. Statistical Analyses

Descriptive statistics were used to describe the collected data, and the average was calculated to verify the central trend and the standard deviation to estimate the existing variability in the data.

T-test was used for anthropometric data comparisons pre- (baseline) and post-intervention. The Shapiro-Wilk test was applied to test the normality of the data. The two-way ANOVA of repeated measures (group × moment) was used to analyze the investigation data, followed by Tukey’s test for multiple comparisons when necessary. The statistical analyses were carried out using the GraphPrism software version 8.0.1 with a significance level of 5% (*p* < 0.05).

## 3. Results

All participants performed an anthropometric assessment before and after the intervention, with no significant differences observed in the variables analyzed (*p* >0.05; Table 1).

Significant differences were observed in the hemodynamic results for SBP, DBP, HR, and DP (Table 2). For SBP, a significant post-intervention reduction was observed only for the RG_A_ (*p* = 0.007) group, and when comparing the RG_A_ and Control groups (*p* < 0.000) after the intervention. For DBP, a significant reduction was observed in the RGA group (*p* = 0.000), and in the head-to-head comparisons of both the RG_A_ (*p* < 0.000) and RG_B_ (*p* = 0.001) groups with the Control Group, after the intervention. Both HR and DP showed significant differences between both the RG_A_ (*p* < 0.0001) and RG_B_ (*p* < 0.000) groups, in head-to-head comparisons with the Control Group, after the intervention.

For both the absolute and relative VO_2max_ (Figure 3 and Figure 4), there was a significant improvement post-intervention in the RG_B_ group (*p* = 0.002 (absolute); *p* < 0.000 (relative)). For the absolute VO_2max_ (Figure 3), RG_B_ performed significantly better than both RG_A_ (*p* = 0.037) and the CG (*p* < 0.000). For the relative VO_2max_ (Figure 4), RG_B_ also performed significantly better than both RG_A_ (*p* < 0.000) and the CG (*p* < 0.000) as measured after the intervention.

For the timed 1600-m walk test (Figure 5), significant reductions in times were observed for both RG_A_ (*p* = 0.027) and RG_B_ (*p* < 0.000). In addition, RG_B_ scored significantly shorter times than both RG_A_ (*p* = 0.000) and the CG (*p* < 0.000) after the intervention.

## 4. Discussion

This study aimed to investigate the effects over time of different forms of neuromuscular training on the cardiovascular and cardiorespiratory capacities of older adults. We observed the effects of two resistance training (RG_A_ and RG_B_) protocols on hemodynamic responses, the estimated VO_2max_, and walking performance in older adults. Comparing the moderate load arm (RG_B_) and the heavier load arm (RG_A_), we see that high load intensities were not necessary to improve cardiorespiratory fitness and walking biomechanics. In fact, several results were significantly superior in the moderate load arm. Globally, both experimental interventions enabled positive results for the individuals tested.

Hemodynamically, significant reductions were observed in SBP, DBP, HR, and DP (Table 2), with RG_A_ and RG_b_ demonstrating superiority compared to the CG after the intervention. For both the absolute and relative VO_2max_ (Figure 3 and Figure 4), a significant improvement between the groups for RG_B_ compared to RG_A_ and the CG was revealed post-intervention. Finally, for the 1600-m walk test (Figure 5), a significant reduction in RG_b_ walk times was found, compared to RG_A_ and the CG after the intervention.

Other studies of resistance training interventions have obtained favorable hemodynamic responses that corroborate our findings. Mota et al. [30] corroborate our hemodynamic findings; for four months, progressive load training and regressive repetitions were performed in hypertensive older women, with significant improvements in SBP (decreasing 14 mm/Hg) and DBP (decreasing 3.6 mm/Hg). Mota et al. [27] applied a progression to each month of intervention loads of 60 to 80% of 1RM and regression in 12 to 8 repetitions. In another study, subjects in Nascimento et al. [31] underwent a 14-week training period and a significant reduction in SBP was shown in hypertensive and normotensive older adults after 10 weeks of training with resistance exercises [32].

Resistance training is the most plausible method for improving functional capabilities and increasing muscle size in older adults [33]. The VO_2max_ was estimated through a 1600-m walk test, considered to be more appropriate than maximum intensity tests, given the restricted functional capacity of many older adults [34]. However, studies on this training modality and oxygen consumption are scarce; therefore, it may be difficult to draw certain conclusions.

However, other studies have shown that resistance training at 80% 1RM alters the cardiac autonomic function and post-exercise maximal oxygen consumption, but not the VO_2max_ kinetics. In the present study, it was observed that the elderly who participated in the RG_B_ group significantly improved the absolute and relative VO_2max_ and had shorter walking times over the pre-intervention period than did the RG_A_ and GC groups; however, the RG_A_ participants improved their VO_2max_, as they obtained a greater supply of O_2_ during the training protocol.

The possible mechanisms producing favorable hemodynamic and cardiorespiratory changes may be related to hormonal, neural, metabolic, and muscular processes [1]. With regard to hemodynamic responses (i.e., HR and BP), better neuro-cardiac adaptations and the renin-angiotensin system may be responsible for positive changes in this system. Additionally, reductions in arterial stiffness, endothelial function, and plasma volume can also occur in response to improved cardiovascular performance [35]. For the VO_2max_, the improved transport and delivery of O_2_ to tissues due to higher hemoglobin and myoglobin are part of the long-term improvement in the VO_2max_ [36]. Another physiological mechanism that can directly influence cardiovascular and respiratory responses is mitochondrial density, which affects muscle cells’ efficiency in absorbing and consuming O_2_ [37]. These changes occur with greater magnitude in aerobic activities, but in resistance training, these changes are also likely. Regardless of the type of activity, the mechanisms activated for possible cardiorespiratory improvements are the same.

The study has some limitations. A direct VO_2_ analysis using the gold standard method (i.e., ergo spirometry) could provide more physiological parameters regarding cardiorespiratory variables (VO_2max_) and metabolic behavior. We used mathematical means to estimate the cardiorespiratory capacity using specific test data (i.e., 1600-m walk test), which also have excellent scientific reproducibility. Another limitation of our study was the scarcity of studies addressing this question. Within this context, most studies investigated the effects of aerobic exercise on the VO_2max_; we found few references to discuss along with our findings. On the other hand, our study showed that resistance training efficiently improves the cardiorespiratory capacity of the elderly. Further studies are needed to bolster and broaden these findings.

## 5. Conclusions

The recognition of the relevance of reflecting on the concept of aging as a focus or reference in public health policies, and their consequent strategies and practices, are presented as a necessary contribution to problematize the real needs of the elderly population. Careful attention to the elderly presupposes providing services with a structure that allows for adequate access and reception, respecting the limitations that the elderly may present. Actions of care through movement with prescriptions for exercises or physical activity planned, prescribed, and performed by a duly qualified professional are the objectives of care actions, which are the main focus of strategies developed in the field of public health.

In our findings, in addition to neuromuscular improvements, resistance training may also show the potential to improve hemodynamic and cardiorespiratory responses in the elderly. Based on the data presented, we can conclude that resistance training at the proposed intensities is an efficient strategy for improving hemodynamic and cardiorespiratory responses and walking performance in the elderly. However, further studies may be needed to investigate the hemodynamic and cardiorespiratory responses associated with resistance training.

In addition to neuromuscular improvements, resistance training shows the potential to improve hemodynamic and cardiorespiratory responses in older adults. Based on the presented data, we can conclude that resistance training at the proposed intensities is an efficient strategy for improving hemodynamic and cardiorespiratory responses and walking performance in the elderly. However, we recognize that more studies may be needed to investigate the hemodynamic and cardiorespiratory responses associated with resistance training.

## Figures and Tables

**Figure 1 ijerph-20-00640-f001:**
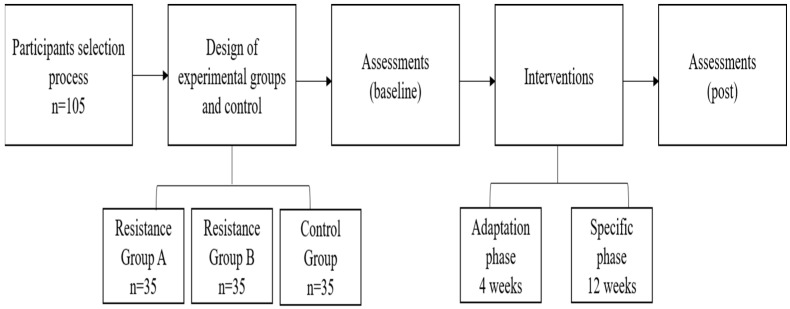
Flowchart of experimental design.

**Figure 2 ijerph-20-00640-f002:**
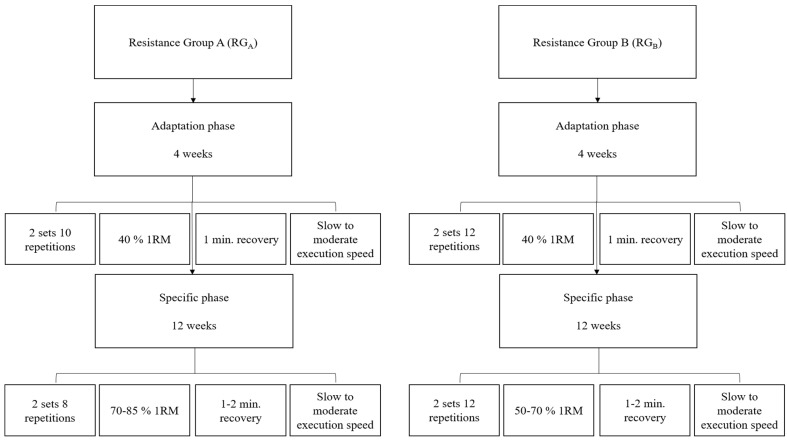
Schematic design of the experimental protocol.

**Figure 3 ijerph-20-00640-f003:**
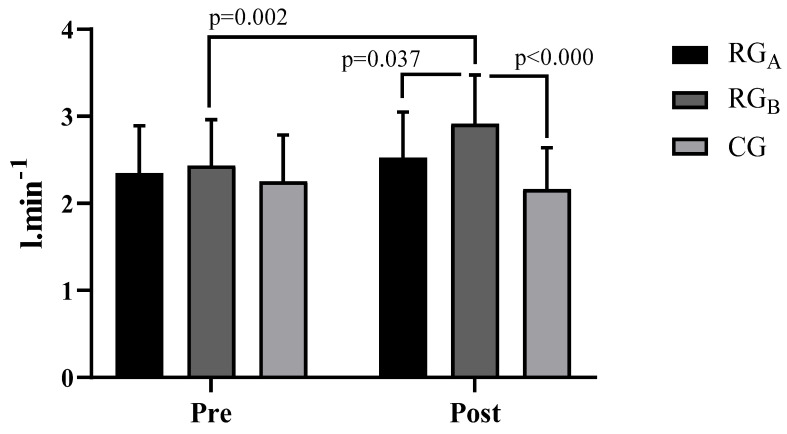
VO_2max_ absolute for the groups before and after the intervention.

**Figure 4 ijerph-20-00640-f004:**
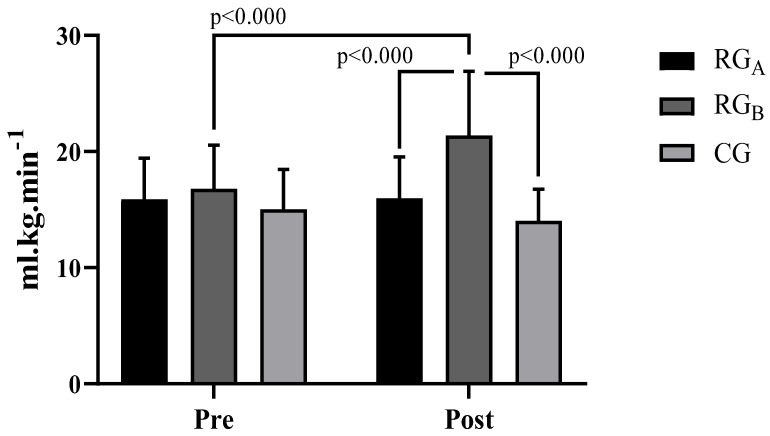
Relative VO_2max_ for the groups before and after the intervention.

**Figure 5 ijerph-20-00640-f005:**
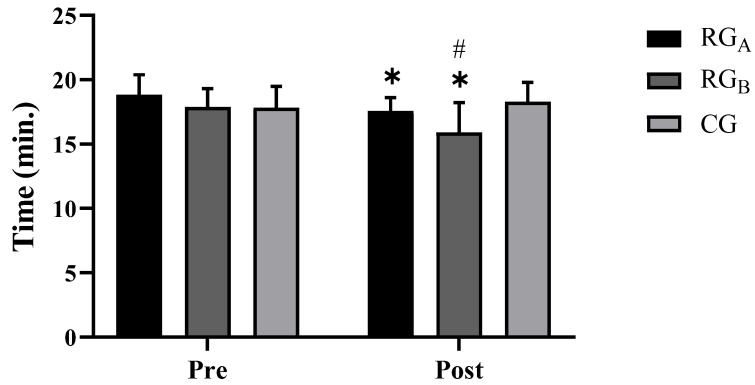
Walking performance for groups, before and after the interventions. * Significant difference compared to the pre-intervention moment for the same group (*p* < 0.05). # Significant difference compared to the strength and Control Group after the intervention (*p* < 0.05).

**Table 1 ijerph-20-00640-t001:** Mean and standard deviation of anthropometric assessment before and after the intervention for training and Control groups.

	Resistance Group A (RG_A_)		Resistance Group B (RG_B_)		Control Group (CG)	
	N = 35 (69.08 ± 5.9 Age)		N = 35 (68.6 ± 5.9 Age)		N = 35 (69.4 ± 8.1 Age)	
Variables	Moment		Moment		Moment	
	Pre	Post	Pre	Post	Pre	Post
Weight(kg)	67.7 ± 11.1	67.7 ± 11.1	66.9 ± 11.9	66.5 ± 11.7	68.6 ± 11.1	70.3 ± 10.6
Height (cm)	158.6 ± 8.9	159.0 ± 9.1	158.2 ± 0.08	158.2 ± 0.08	163.6 ± 0.07	163.6 ± 0.07
BMI (kg/m^2^)	27.03 ± 4.04	26.8 ± 4.1	26.6 ± 3.7	26.6 ± 3.7	26.08 ± 4.5	26.4 ± 4.3

**Table 2 ijerph-20-00640-t002:** Hemodynamic assessment before and after the intervention for training and Control groups.

Variables	Groups					
	Resistance Group A		Resistance Group B		Control Group	
	Moment		Moment		Moment	
	Pre	Post	Pre	Post	Pre	Post
SBP (mm/Hg)	136.57 ± 17.14	122.25 ± 12.83 *^†^	142.28 ± 18.32	132.85 ± 14.46	141.14 ± 17.95	142.57 ± 18.04
DBP (mm/Hg)	85.42 ± 10.93	76.12 ± 8.43 *^†^	81.42 ± 6.01^†^	79.42 ± 6.39	82.28 ± 7.70	87.42 ± 9.50
HR (bpm)	86.97 ± 17.07	92.64 ± 9.69 ^†^	97.48 ± 10.67	93.60 ± 9.36 ^†^	105.82 ± 10	107.88 ± 9.48
DP (bpm × mm/Hg)	11,862.28 ± 2658.19	11,312.25 ± 1604.50 ^†^	13,886.85 ± 2459.58	12,429.71 ± 1808.27 ^†^	14,923.42 ± 2291.63	14,357.71 ± 2164.50

* Significant difference compared to the intra-group pre-intervention moment (*p* < 0.05); ^†^ Significant difference compared with the Control Group for the post-intervention moment (*p* < 0.05). SBP: Systolic Blood Pressure; DBP: Diastolic Blood Pressure; HR: Heart Rate; DP: Double Product.

## Data Availability

The datasets analyzed during the current study are available from the first author [SCARTONI, FR] upon immediate request.

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
