# Peer review of "Chronic Effects of Different Types of Neuromuscular Training on Hemodynamic Responses Estimated VO2max, and Walking Performance in Older People"

_ijerph, 2022, doi:10.3390/ijerph20010640_

Round 1
Reviewer 1 Report
Dear authors, the present article does not reach the required quality standard of this journal.
Despite the subject is intersting, the originality is weak considering the available data in the literature. Please see the following reference : Smart, T. F. F., Doleman, B., Hatt, J., Paul, M., Toft, S., Lund, J. N., & Phillips, B. E. (2022). The role of resistance exercise training for improving cardiorespiratory fitness in healthy older adults: a systematic review and meta-analysis. Age and ageing, 51(6), afac143. https://doi.org/10.1093/ageing/afac143
In addition, there are a lot of mistakes in the text.
Example 1 : see lines 57-58 ("However, we must better comprehend this popu- 57 lation’s neuromuscular training on cardiorespiratory capacity"). A proofreading by an English native speaker is necessary.
Example 2: The missing information concerning the signification of the symbols in the legend of the Table 2 does not allow the reader to understand the comparisons.
Example 3: see lines 181-182 : the HR was increased in group A and decreased in group B.
Moreover, it is not clear whether the control group did not really have any training during the 16 weeks. How the authors controlled that important point ?
Finally, the differences observed between group A and group B are very light and not really significant on a physiological base. This result is certainly due to the fact that both training are nearly the same.
Reviewer 2 Report
Abstract: Please check abstract style, it should not be written in separate section, objectives, method sresults...
Meterial and methods: what are the data collection standard! to collect anthropometrics data, how to calibration it.
Results: For presenting the data in table 1 , table2,and figures, what statistics should be added to test the difference!
Dicussion : the control group shoold be discussed after the trial closure of what activities the researcher performed.
References: please check reference style and typing error ex. no. 1, 6,10,13,18...
Author Response
Summary: Check the style of the summary; it should not be written in separate sections,
objectives, or results of the method...
Verified summary
Material and methods: what are the data collection standards? Collect anthropometric
data how to calibrate them.
AS DESCRIBED IN THE LINES 126-141
Anthropometric Analyzes
Body composition was measured using two parameters: body mass index and left calf
circumference [26]. Body mass and height are measured to verify the Body Mass Index, which is
essential for assessing the nutritional and anthropometric profile of older adults [27].
In the protocol to measure body mass and height, the participant is weighed barefoot and in
exercise clothes (light clothes, shorts, and shirt), in a standing position, in the central part of the
platform of the Filizola® brand mechanical scale (Brazil) with INMETRO seal, accurate to 100 g,
with weight measured in kilograms.
To measure height, the same standard mentioned above is used, with an aluminum
stadiometer, with 1mm precision, standing upright, arms extended along the body, feet together,
in inspiratory apnea, with the head oriented in the Frankfurt plane. Height was measured in
centimeters [28].
Then, measurements of body mass and height were used to calculate the body mass index
(BMI), obtained by the ratio between the measurements expressed in the formula below, with
body mass being measured in kilograms (kg) and height measured in meters (m): BMI = Body
mass/height in m2
.
Results: To present the data in Tables 1, 2, and Figures, what statistics should be added
to test the difference?
AS DESCRIBED IN THE LINES 166-168
Descriptive statistics were used to describe the collected data, where the average was
calculated to verify the central tendency and the standard deviation to estimate the existing
variability in the data.
Discussion: the control group should be discussed after the end of the trial about what
activities the researcher performed.
LINES 99-102 The control group (CG) maintained routine daily tasks throughout the study. In
addition, this group did not perform any systematic physical activity during the 16 weeks. Once
a week, the control group met with the evaluators to report their daily activities routine for the
week.
Referências: verifique o estilo de referência e o erro de digitação, por exemplo. no. 1, 6,
10, 13, 18...
DONE

Reviewer 3 Report
Dear authors,
I attached the manuscript with some considerations. The most important is that there is not statistical analyses in order to check if the three groups are equal or different in the baseline.
On the other hand, it is important to rewrite the intervention in both groups, because some information is not clear.
Finally, there are another minor considerations.

Author Response
Dear reviewer, all changes were accepted and made. The comments are in the PDF.
Round 2
Reviewer 1 Report
Dear authors, unfortunately the paper in its present does not bring enough novelty in the field of training in older subjects.
As a consequence, It cannot be consider for publication in IJERPH.
Best regards.
Author Response
The manuscript submitted to Sport and Health, in its special volume, Positive
Effects of Leisure & dash, Sports and Physical Activity on Public Health, presents an extremely important issue for training older adults. Since it is part of the primary motor skills of The human being undergoes changes in the mechanics of movement with the aging process, often attributed to a decrease in muscle strength and cardiorespiratory provision. Many studies point to neuromuscular training as a variable for improving cardiorespiratory restriction, which is not new, as presented by the reviewer in the reference below
Smart, T.F.F., Doleman, B., Hatt, J., Paul, M., Toft, S., Lund, J.N., & Phillips, B.E. (2022). The role of resistance training in improving cardiorespiratory fitness in healthy older adults: a systematic review and meta-analysis. Age and Aging, 51(6), afac143. https://doi.org/10.1093/ageing/afac143
However, the novelty of our study is based not only on strength training and the
improvement of maximum VO2 but on its practical applicability as a function of the intensities to be prescribed. In the field of action, it always raises doubts about whether the most effective training should be low to moderate or high intensity. Concerning walking performance, it becomes the possession of this knowledge, as the correct prescription of intensity protects the cardiovascular system and optimizes the functions of the locomotor system. So, I believe our study brings news in training science, leisure, and public health.
It should be noted that walking is the modality most practiced by the elderly
population and the most democratic in the field of practices, in addition to providing autonomy and independence of movement for this population.

Reviewer 3 Report
Dear authors,
Thank you for accepting the changes. However, I attach the document in order to revise some minor considerations.
Thank you.

Author Response
(I)Please check that all references are relevant to the contents of the Manuscript.
All references were checked.
IN LINE 91, the reference [23] was removed from the references. It wasn't necessary "To measure muscle strength, the maximum weight test of one repetition maximum (1RM) was conducted for [23] the following exercises: bench press, simultaneous squat, one-arm dumbbell row, seated knee extension, simultaneous standing biceps, standing knee flexion, and triceps on the bench".
(II) Any revisions to the manuscript should be marked up using the "Track
Changes" function if you are using MS Word/LaTeX, such that any changes can
be easily viewed by editors and reviewers.
Done. All modification requests are highlighted in the text and explained below
The questions below appear in order as requested by the reviewer.
• FOR THE QUESTION: Which was the hypothesis?
The hypothesis is: neuromuscular training for both low to moderate and high intensities improves VO2 max and walking performance
• IN LINE 91: the reference was removed. It wasn't necessary
To measure muscle strength, the maximum weight test of one repetition maximum (1RM) was conducted for [23] the following exercises: bench press, simultaneous squat, one-arm dumbbell row, seated knee extension, simultaneous standing biceps, standing knee flexion, and triceps on the bench.
• Change the font or type because it is the same as the text.
Done in the text
• It is important to add information about the significant differences and both
symbols used as a note for the table.
* Significant difference compared to the intra-group pre-intervention moment (p<0.05);
† Significant difference compared with the Control Group for the post-intervention moment (p<0.05). SBP: Systolic Blood Pressure; DBP: Diastolic Blood Pressure; HR: Heart Rate; DP: Double Product
• These two paragraphs must be written together because a paragraph with only
one sentence is incorrect.
However, other studies have shown that resistance training at 80% of 1RM alters
the cardiac autonomic function and maximal oxygen consumption after exercise but not VO2 max kinetics in young individuals, thus reflecting long-term physiologic change [40].
However, our study observed that older adults who participated in RGB
improved significantly in absolute and relative VO2 max. Consequently, the RGB group also showed a shorter execution time in pre-intervention than RGA and CG in postintervention time; however, RGA participants improved VO2 max since they had a higher O2 supply during the training protocol.
IT WAS REPLACED BY
However, other studies have shown that resistance training at 80% 1RM alters
the cardiac autonomic function and post-exercise maximal oxygen consumption but not VO2 max kinetics. In the present study, it was observed that the elderly who participated in the RGB significantly improved the absolute and relative VO2max and shorter walking time in the pre-intervention period than the RGA and GC in the postintervention period; however, the RGA participants improved their VO2 max, as they obtained a greater supply of O2 during the training protocol.
